# Lipid Membrane Remodeling by the Micellar Aggregation of Long-Chain Unsaturated Fatty Acids for Sustainable Antimicrobial Strategies

**DOI:** 10.3390/ijms24119639

**Published:** 2023-06-01

**Authors:** Sungmin Shin, Hyunhyuk Tae, Soohyun Park, Nam-Joon Cho

**Affiliations:** School of Materials Science and Engineering, Nanyang Technological University, 50 Nanyang Avenue, Singapore 639798, Singapore; sungmin001@e.ntu.edu.sg (S.S.); hyunhyuk001@e.ntu.edu.sg (H.T.); soohyun.park@ntu.edu.sg (S.P.)

**Keywords:** antimicrobial lipid, long-chain unsaturated fatty acid, antibacterial, lipid bilayer membrane, quartz crystal microbalance–dissipation, model membrane system

## Abstract

Antimicrobial fatty acids derived from natural sources and renewable feedstocks are promising surface-active substances with a wide range of applications. Their ability to target bacterial membrane in multiple mechanisms offers a promising antimicrobial approach for combating bacterial infections and preventing the development of drug-resistant strains, and it provides a sustainable strategy that aligns with growing environmental awareness compared to their synthetic counterparts. However, the interaction and destabilization of bacterial cell membranes by these amphiphilic compounds are not yet fully understood. Here, we investigated the concentration-dependent and time-dependent membrane interaction between long-chain unsaturated fatty acids—linolenic acid (LNA, C18:3), linoleic (LLA, C18:2), and oleic acid (OA, C18:1)—and the supported lipid bilayers (SLBs) using quartz crystal microbalance-dissipation (QCM-D) and fluorescence microscopy. We first determined the critical micelle concentration (CMC) of each compound using a fluorescence spectrophotometer and monitored the membrane interaction in real time following fatty acid treatment, whereby all micellar fatty acids elicited membrane-active behavior primarily above their respective CMC values. Specifically, LNA and LLA, which have higher degrees of unsaturation and CMC values of 160 µM and 60 µM, respectively, caused significant changes in the membrane with net |Δ*f*| shifts of 23.2 ± 0.8 Hz and 21.4 ± 0.6 Hz and Δ*D* shifts of 5.2 ± 0.5 × 10^−6^ and 7.4 ± 0.5 × 10^−6^. On the other hand, OA, with the lowest unsaturation degree and CMC value of 20 µM, produced relatively less membrane change with a net |Δ*f*| shift of 14.6 ± 2.2 Hz and Δ*D* shift of 8.8 ± 0.2 × 10^−6^. Both LNA and LLA required higher concentrations than OA to initiate membrane remodeling as their CMC values increased with the degree of unsaturation. Upon incubating with fluorescence-labeled model membranes, the fatty acids induced tubular morphological changes at concentrations above CMC. Taken together, our findings highlight the critical role of self-aggregation properties and the degree of unsaturated bonds in unsaturated long-chain fatty acids upon modulating membrane destabilization, suggesting potential applications in developing sustainable and effective antimicrobial strategies.

## 1. Introduction

Antimicrobial fatty acids have been widely applied in various applications, such as pharmaceutical applications, cosmetic applications, detergents, food science, and nanotechnology [1,2,3,4,5,6,7]. As the global demand for surfactants has increased, the surfactant market is expected to grow at a compound annual growth rate (CAGR) of 4.5% between 2022 and 2030 [3]. However, most surfactant-based products are principally derived from petroleum via a chemical process and leave residues that can be detrimental to the environment [3,8]. Due to the increasing market demand for environmentally friendly and safer ways to produce surfactants, the surfactant industry has shifted away from using synthetic surfactants to replace them with more sustainable alternatives directly derived from natural fatty acids, which are also known as green surfactants or biosurfactants [1,8,9,10].

Compared to their synthetic counterparts, natural fatty acids have superior physicochemical properties, such as a lower critical micelle concentration and high toleration relative to high temperatures as well as superior pH and ionic strength [1,11,12,13,14]. Moreover, fatty acids can target bacterial cell membranes in multiple mechanisms, which can lead to the inhibition of bacterial growth or cell death [15]. They accomplish this by destabilizing the lipid membrane via their amphiphilic properties, which disrupts metabolic regulation and prevents bacterial growth [15,16]. Some of them can even cause complete membrane lysis, resulting in irreversible damage and leading to bacterial death in a matter of minutes [17]. As a result, natural fatty acids offer a promising antimicrobial approach for combating bacterial infections and preventing the development of drug-resistant strains [2,16,18], and they offer a sustainable strategy and increase environmental awareness [1,3,8,9].

Among various natural fatty acids, long-chain unsaturated fatty acids have been classified as generally recognized as safe (GRAS) according to the US Food and Drug Administration’s (FDA) classification system [19] and are widely used as commercial formulations in food, medicine, cosmetic, and agriculture industries [2,20,21,22,23]. In addition to their potential as antimicrobial agents against a wide range of bacteria, unsaturated fatty acids have been found to be more effective than saturated ones with the same carbon chain [16,24,25]. In line with thoughts that antimicrobial lipids are principally active in the micellar state [26], their antimicrobial activities may be linked to their ability to form micelles or aggregates in a solution. However, although extensive in vitro studies revealed the structural effect of long-chain unsaturated fatty acids on antibacterial activity [16], it remains unknown how they interact and destabilize membranes in real time.

Herein, we systematically investigated the molecular self-assembly and membrane interaction of long-chain unsaturated fatty acids—linolenic acid (LNA, C18:3), linoleic (LLA, C18:2), and oleic acid (OA, C18:1)—on supported lipid bilayer (SLB) platforms in real time (Figure 1). SLB platforms were selected as model cell membranes because they are well suited for accessing the interfacial activity of membrane-active molecules, and they are compatible with a diverse array of surface-sensitive methods. We first investigated the molecular aggregation of these compounds at various concentrations using fluorescence spectroscopy. Then, the real-time membrane remodeling by long-chain unsaturated fatty acids was monitored with quartz crystal microbalance-dissipation (QCM-D). Finally, time-lapsed fluorescence microscopy imaging techniques were performed to scrutinize membrane morphological responses between long-chain unsaturated fatty acids and model cell membranes.

## 2. Results and Discussion

### 2.1. Determination of the Critical Micelle Concentration

We first measured the critical micelle concentration (CMC) values of LNA, LLA, and OA, as single-chain antimicrobial lipids (e.g., fatty acids) are suggested to disrupt the phospholipid membrane by forming a micellar structure [26]. To determine CMC values, fluorescence spectroscopy experiments were conducted with the fluorescent probe technique by measuring the fluorescence emission spectrum of 1-pyrenecarboxaldehyde in different concentrations of the test compounds. When present in an aqueous environment, 1-pyrenecarboxaldehyde exhibits a peak emission wavelength of 473 nm, and in the presence of micelle aggregates, the peak wavelength decreases as a result of the probe intercalating into the hydrophobic region of the micelle, which corresponds to a decrease in dielectric constant [30,31]. Thus, CMC is defined as the lowest concentration at which the peak wavelength begins to decrease, and the formation of micelles becomes energetically favorable. The CMC experiments were performed under a typical physiological value of pH 7.5. The CMC value of LNA in PBS was determined to be 160 μM, which is consistent with the literature value [32] (Figure 2A). Similarly, the CMC of LLA was determined to be 60 μM, which also agrees with the literature value [33] (Figure 2B). Finally, the CMC value of OA was found to be 20 μM, which agrees with the value reported in the literature [32,33] (Figure 2C). Together, lower CMC values were observed when there were few unsaturated bonds. This trend in CMC values can be attributed to the lower hydrophobicity of the long-chain fatty acid with more C=C bonds. When one -CH_2_- group is lost, it results in the formation of one double bond, which reduces hydrophobicity and increases solubility, leading to higher CMC values [14]. Taken together, the results indicate that the unsaturation degree affects the self-assembly of fatty acids; therefore, LNA has a higher CMC value than other long-chain unsaturated fatty acids.

### 2.2. Interaction between Long-Chain Unsaturated Fatty Acids and Supported Lipid Bilayers

We then performed QCM-D experiments to probe the membrane remodeling behavior of LNA, LLA, and OA against zwitterionic DOPC supported lipid bilayers (SLBs) under physiological conditions. The QCM-D technique measures the changes in frequency (Δ*f*) and energy dissipation (Δ*D*) signals that occur due to the mechanistic interaction between SLBs and antimicrobial lipids that are reflected as mass and viscoelastic properties. Zwitterionic DOPC lipid compositions were used to fabricate the SLB platform for two reasons. First, phosphocholine lipids are widely found in biological membranes, and second, the lipid composition has demonstrated distinct membrane morphological responses in previous studies [34,35,36,37,38,39]. Single-component SLBs with zwitterionic DOPC lipids were formed by the solvent-assisted lipid bilayer (SALB) method [40]. As for complete SLB formation, an initial baseline recording was obtained in an aqueous buffer solution, after which the solution was exchanged for a water-miscible isopropanol solution. Next, 0.5 mg/mL of DOPC lipids in isopropanol was deposited on the silicon-dioxide-coated sensor’s surface. The solution was then exchanged with the aqueous buffer solution. The homogenous SLB formation was confirmed by the final resonance frequency (Δ*f*) and energy dissipation (Δ*D*) shifts of −26.4 ± 0.6 Hz and 0.3 ± 0.1 × 10^−6^, respectively, as reported in the literature [41,42]. The bilayer was further confirmed to be defect-free by the bovine serum albumin (BSA) blocking step, which resulted in a negligible change in frequency of less than 1 Hz and a significant reduction in protein adsorption relative to the SLB-coated silica substrate by over 94%. Subsequently, SLBs were exposed to LNA, LLA, and OA at fixed concentrations under continuous flow conditions, followed by a washing step with an equivalent buffer solution, and QCM-D measurements for Δ*f* and Δ*D* shifts were tracked in real time. Of note, t = 0 min denotes the SLB formation, and t = 5 min indicates the addition of test compounds under continuous flow conditions in the measurement chamber. The binding dynamics obtained for the test compounds are presented below.

Linolenic Acid (LNA): Figure 3 presents the effects of LNA on the membrane remodeling of SLBs. Upon treatment with 500 μM LNA, there was a rapid decrease in Δ*f* to around −39.2 ± 1.8 Hz and an increase in Δ*D* to 8.6 ± 0.2 × 10^−6^ (Figure 3A). The increase in mass and dissipation showed that LNA was attached to the SLB right after the treatment. Upon reaching the critical point, measurement responses began to reverse, with a swift rise in Δ*f* and decline in Δ*D*, ultimately stabilizing at −5.3 ± 2.5 Hz and 6.3 ± 0.4 × 10^−6^, respectively. The rapid increase in frequency and large residual dissipation shift may suggest that the LNA treatment had partially destabilized the lipid membrane. Interestingly, a buffer washing step led to a decrease in Δ*f* shifts to around −31.0 ± 2.4 Hz and Δ*D* shifts to 0.2 ± 0.1 × 10^−6^. This intricate behavior parallels the capric acid treatment on SLBs at high concentrations in which similar QCM-D signals were observed for the lipid–membrane interaction with a net Δ*f* and Δ*D* decrease, but the behavior contrasts with lauric acid treatments on SLBs at concentrations above CMC with respect to a net Δ*f* and Δ*D* increase [37,38]. LNA produced a similar activity profile at a concentration of 250 μM (Figure 3B). The Δ*f* signal gradually decreased to −38.1 ± 3.4 Hz and then increased before finally stabilizing at −28.0 ± 1.7 Hz. At the same time, the signal Δ*D* followed the same pattern, reaching a critical point at 12.2 ± 0.5 × 10^−6^ before decreasing and stabilizing at 7.9 ± 0.3 × 10^−6^. During the buffer washing step, there was a sharp decrease in both Δ*f* and Δ*D* signals, with final values of −28.0 ± 2.1 Hz and 0.3 ± 0.4 × 10^−6^, respectively. When treated with LLA at lower concentrations (125 μM and below), there were either insignificant or minimal changes in both Δ*f* and Δ*D* signals (Figure 3C–F). This observation is consistent with previous reports in which the micellar form of lipids facilitates membrane disruption [35]. Collectively, the QCM-D findings suggest that LNA micelles are active against SLBs, whereas LNA monomers are primarily inactive against SLBs at concentrations below the CMC value (160 μM).

Linoleic Acid (LLA): Figure 4 displays the effects of LLA on the membrane remodeling of SLBs. The treatment with 500 μM LLA resulted in a rapid reduction in Δ*f* shift to −39.3 ± 2.7 Hz and an increase in Δ*D* shift to 9.8 ± 0.4 × 10^−6^ (Figure 4A). The Δ*f* signal subsequently began to increase to −11.0 ± 2.0 Hz, and the Δ*D* signal reached around 4.5 ± 0.2 × 10^−6^ before gradually increasing. After a buffer washing step, the Δ*f* measurement began to increase with a decrease in Δ*D* and stabilized at around −1.5 ± 1.0 Hz and 5.1 ± 0.4 × 10^−6^. Similar results were observed when SLBs were treated with 250 μM LLA, with values of around −4.5 ± 0.9 Hz and 4.2 ± 0.6 × 10^−6^ (Figure 4B). Notably, unlike LNA, no significant alterations were noticed in Δ*f* and Δ*D* shifts after buffer washing for up to 140 min at 500 uM and 250 uM of the LLA treatment. Due to LLA having a straighter hydrophobic tail compared to LNA, LNA may not bind as strongly to the membrane as LLA [16,43]. Conversely, rapid changes were measured in Δ*f* and Δ*D* shifts following buffer washing at 125 uM of LLA treatment, leading to a final Δ*f* shift value of −31Hz and Δ*D* shift of 1.62 × 10^−6^ (Figure 4C). The same trend was detected at 63 uM of LLA following buffer washing, even though the changes in Δ*f* and Δ*D* shifts transpired faster than in 125 uM of LLA treatment (Figure 4D). It is noteworthy that LLA had a slow binding rate relative to the membrane at lower concentrations, and the changes in Δ*f* and Δ*D* shifts occurred rapidly after buffer washing. This result indicates that the binding strength between LLA micelles and the membrane diminishes as the concentration of LLA decreases [44]. Similarly to the activity profile of LNA, there were negligible changes in both measurement responses at lower concentrations (31 μM and 16 μM, Figure 4E,F). Thus, LLA was not active against SLBs at concentrations below a CMC value of 60 μM.

Oleic Acid (OA): The effects of OA on the remodeling of SLBs are presented in Figure 5. Upon treatment with 500 μM OA, there was an immediate decrease in Δ*f* shift to −43 Hz and an increase in Δ*D* shift to 11.3 × 10^−6^ (Figure 5A). Subsequently, the Δ*f* signal increased to around −12.7 ± 0.4 Hz, while the Δ*D* signal decreased to around 9.8 ± 1.2 × 10^−6^ and gradually decreased thereafter. After buffer washing, the Δ*f* signal began to increase to −2.1 ± 0.2 Hz, and Δ*D* began to decrease to 3.4 ± 0.9× 10^−6^. Similar profiles were observed at 250 μM OA, with final Δ*f* and Δ*D* values around −1.3 ± 0.2 Hz and 4.6 ± 0.5 × 10^−6^ (Figure 5B), respectively. Interestingly, OA causes membrane remodeling at a slower rate than LNA and LLA at equivalent concentrations. It was observed that 125 μM of OA induced slow membrane remodeling on SLB compared to higher OA concentrations (Figure 5C). A similar trend was observed after treatment with 63 μM OA (Figure 5D). In addition, 31 μM of OA did not cause substantial changes in membrane mass and viscoelasticity, although the concentration was above CMC (Figure 5E). Instead, the OA micelles moderately attached to the SLB, leading to a gradual decrease in Δ*f* shift and an increase in Δ*D* shift, even after buffer washing. Even though low concentrations of OA micelles can trigger membrane remodeling due to their lower CMC value compared to LNA and LLA, the impact of OA on the alteration of the membrane’s structure was not as significant as that of LNA and LLA. This is likely due to the fact that OA’s unsaturated tails have a limited degree of conformational flexibility in their shape, resulting in more rigid packing and reduced fluidity in the membrane [45]. When SLBs were treated with an OA monomer below CMC, there was a negligible change in Δ*f* and Δ*D* shifts (Figure 5F), demonstrating that the OA monomer does not induce membrane destabilization.

### 2.3. Trend in the Interaction Kinetics of Long-Chain Unsaturated Fatty Acids on Supported Lipid Bilayers

To better understand the membrane-active mechanism of long-chain unsaturated fatty acids, we examined the trends in the interaction between three fatty acids (LNA, LLA, and OA) and SLBs at concentrations above and below their respective CMC values. Treatment with 250–500 μM LNA resulted in rapid binding and membrane remodeling (Figure 6A). LLA showed similar response profiles above CMC, but the interaction kinetics slowed down at lower concentrations (Figure 6B). OA exhibited slower kinetics compared to LNA and LLA across all concentrations (Figure 6C). At high concentrations above CMC, it appears that long-chain fatty acids with a higher degree of unsaturation require fewer molecules to drive the system to a non-lamellar phase and induce curvature, possibly due to the increased kink structure and the greater intermolecular distance between fatty acid molecules [46]. The double-bond-induced negative curvature increases concentrations dependently, and this leads to an increase in bending rigidity, which is determined by the trade-off between Born energy and hydrophobic interaction energy [14]. Conversely, when the concentrations of fatty acids are below their respective CMC, in which they are predominantly in the monomer state, none of them induce membrane remodeling with negligible change in Δ*f* and Δ*D* shifts compared to those induced by fatty acids above their CMC values (Figure 6D–F). This highlights the importance of micelle formations for the antimicrobial membrane-active functionality of fatty acids. It is noteworthy that long-chain unsaturated fatty acids with a higher degree of unsaturation not only led to significant changes in the membrane but also required high concentrations to initiate membrane remodeling, as CMC values are generally proportional to the degree of unsaturation. Moreover, it is important to note that the concentrations above CMC align with the minimum inhibitory concentration (MIC) values reported for these fatty acids: 200–400 μM for LNA against *Staphylococcus aureus* and *Staphylococcus pyrogenes* [47], 114–125 μM for LLA against *Bacillus cereus* [48]*,* and 53 μM for OA against *Porphyromonas gingivalis* [49]. Although MIC values may vary in the literature due to different experimental conditions, it can be inferred that long-chain unsaturated fatty acids primarily induce concentration-dependent membrane remodeling behavior in vitro, particularly above their respective CMC values. Taken together, the observed trends support the following: the degree of unsaturation of long-chain fatty acids has a significant effect on self-assembly properties and differential membrane remodeling behavior on phospholipid membranes.

### 2.4. Observation of Membrane Morphological Responses in Supported Lipid Bilayers

To further characterize membrane remodeling, we performed time-lapse fluorescence microscopy to directly observe the response in membrane morphology induced by fatty acids. The SLBs, which comprise 99.5 mol % DOPC and 0.5 mol % Rh-PE, were fabricated on a hydrophilic silicon dioxide surface using the SALB method at a pH of 7.5. Once the baseline signal was established to signify bilayer formation, the test compounds in an equivalent PBS buffer solution were introduced under continuous flow conditions. Of note, t = 0 min corresponds to the time when the solution containing the test compounds reached the measurement chamber. Based on fluorescence spectroscopy and QCM-D measurement results, we tested LNA, LLA, and OA at one concentration above CMC and one below CMC. The selected concentrations were 500 μM and 63 μM for LNA, 250 μM and 31 μM for LLA, and 125 μM and 16 μM for OA.

Linolenic acid (LNA): Figure 7A presents the time-lapsed morphological responses in the SLB, which were induced by 500 μM LNA (above CMC). Within 2.8 min of the treatment, a profuse number of short tubules protruded from the SLBs, in which bright spots represent the nucleation sites. Shortly after, within 11.4 min, the tubules started to aggregate at higher fluorophore concentrations, and small dark patches started to emerge in the background. As the time scale reached the final mark, aggregated tubules became more prominent when the fluorescence intensity of the background decreased significantly. This discrepancy may be attributed to how the tubules grow out of the focal plane [50]. Upon buffer washing, the tubules were removed, and variously sized dark patches appeared with very few bright spots of fluorescence, indicating that LNA induced membrane disruption in the SLB membrane. In marked contrast, minimal morphological responses were observed, with only a few long tubules protruding from the SLB when treated with 63 μM LNA (Figure 7B). Upon buffer washing, most tubules were removed with negligible morphological changes in the SLB, which agrees with QCM-D results.

Linoleic acid (LLA). Figure 8A shows the morphological change in the SLB over time when treated with 250 μM LLA under continuous flow conditions. Initially, many elongated tubules protruded from the SLBs within 5.6 min, and shortly after, the tubules started to lose their fibrils and aggregated with small dark patches emerging from the SLB. As time passed, aggregated tubules with fibrils (approximately 2 μm in length) became more prominent, and the fluorescence intensity of the background decreased significantly, similarly to LNA treatments. Interestingly, compared to LNA treatments, the number of nucleation sites (indicated by bright spots) was lower, and the size of tubules was larger when treated with LLA. This implies that a higher amount of LLA is required to attach to the membrane in order to induce morphological remodeling, resulting in large lipid aggregations. These findings are consistent with the interaction kinetics of fatty acids measured in QCM-D. Upon buffer washing, the tubules were removed, and various sizes of dark patches with very few bright spots of fluorescence appeared. It appears that LLA induced membrane remodeling in the SLB membrane. Figure 8B presents the morphological responses in an SLB upon treatment with 31 μM LLA (below CMC). In this case, a relatively smaller number of tubules protruded from the SLB, and a decrease in fluorescence intensity in the background was not observed, indicating that there were fewer membrane morphological changes compared to the LLA treatments above CMC. After buffer washing, most aggregates were removed from SLBs, which is in agreement with QCM-D results.

Oleic acid (OA): Figure 9A presents time-lapsed snapshots of SLB’s morphological response induced by 125 μM OA (above CMC). Within a few minutes, elongated tubules began to form and varied in length between 2 and 30 μm. The tubules remained stagnant until 14.6 min when the fibrils were absorbed, and this led to an increase in the tubule’s size, which may be a result of potential aggregation. Of note, the tubules formed by OA were larger than those induced by LNA and LLA, suggesting that the decreased kinked structure of OA requires large aggregations for membrane remodeling. Upon the buffer washing step, the tubules remained attached to the lipid bilayer but decreased in size. It appears that the highest saturation degree and hydrophobicity of OA resulted in strong insertions in the membrane. At 16 μM OA (below CMC), dense coverage of tubules with fibrils protruded from the SLB after OA treatments (Figure 9B). Upon buffer washing, most tubules were removed, but a few bright spots of tubules that were larger than the tubules produced by 31 μM LLA remained translocated in the SLB. This may suggest that due to the smaller kinked molecular structure of OA, the packing between phospholipids increases and decreases the membrane’s fluidity, resulting in larger tubules that are adsorbed in the membrane [51,52].

## 3. Materials and Methods

### 3.1. Materials

1,2-Dioeloyl-sn-glycero-3-phosphocholine (DOPC) and 1,2-dioleoyl-sn-glycero-3-phosphoethanolamine-N-(lissamine rhodamine B sulfonyl) (ammonium salt) (Rh-PE) were purchased from Avanti Polar Lipids, Inc. (Alabaster, AL, USA). 1-Pyrenecarboxaldehyde was purchased from Sigma-Aldrich (St. Louis, MO, USA). Linolenic acid, linoleic acid, and oleic acid were obtained from Sigma-Aldrich (St. Louis, MO, USA). Phosphate-buffered saline (PBS) was procured from Gibco (Carlsbad, CA, USA). All solutions were prepared using Milli-Q-treated deionized water (>18 MΩ·cm) (Millipore, Billerica, MA, USA).

### 3.2. Preparation of Fatty Acid Solutions

Stock solutions of linolenic acid (LNA), linoleic acid (LLA), and oleic acid (OA) were prepared by dissolving the weighed amount of the compound in ethanol to a concentration of 50 mM. Aliquots of the stock solution were diluted 100-fold with a PBS solution to a final concentration of 500 μM. Complete solubilization was promoted by heating the test samples to 70 °C for 15 min. Subsequently, the solutions were cooled to room temperature and further diluted in 2-fold steps. Linolenic acid (p*K*a ~8), linoleic acid (p*K*a ~9), and oleic acid (p*K*a ~10) are all assumed to be protonated under all test conditions. All solutions were prepared immediately before the experiment.

### 3.3. Fluorescence Spectroscopy

Experiments were conducted using a Spark^®^ Multimode Microplate Reader (Tecan Trading AG, Männedorf, Switzerland) to determine the critical micelle concentration (CMC) values of the tested compounds at room temperature (21 °C). The test samples were excited at 365.6 nm, and the fluorescence emission spectrum of the probe in PBS was recorded from 400 nm to 600 nm in the presence of increasing concentrations of the samples [53]. The stock solution of the probe was initially prepared in methanol at a concentration of 5 mM. The test samples were prepared by adding a certain amount of the probe stock to a glass vial, and then methanol was fully evaporated using nitrogen gas. A PBS solution containing an appropriate amount of test compound was then added to the vial, followed by vortexing and nitrogen gas evaporation. The final concentration of the probe was 0.1 μM. All measurements for each sample were scanned ten times and averaged.

### 3.4. Quartz Crystal Microbalance–Dissipation (QCM-D)

QCM-D experiments were conducted to characterize the SLB formation process and the interaction between antimicrobial lipids and the SLB using a four-channel Q-Sense E4 instrument (Q-Sense AB, Gothenburg, Sweden). The QCM-D technique monitors shifts in the frequency (Δ*f*) and energy dissipation (Δ*D*) of an oscillating, piezoelectric quartz crystal sensor chip as a function of time, and these shifts reflect the mass and viscoelastic properties, respectively, of the adsorbed phospholipid film on the surface [54]. The sensor chips had a fundamental frequency of 5 MHz [54,55,56] and were coated with a sputter-coated, 50 nm thick layer of silicon dioxide (model no. QSX 303, Biolin Scientific AB, Västra Frölunda, Switzerland). For cleaning purposes, the sensor chips were sequentially washed with SDS 1% (*w*/*v*), water, and ethanol and then dried with a gentle stream of nitrogen gas, followed by oxygen plasma treatment for 1 min with an Expanded Plasma Cleaner (PDC-002, Harrick Plasma, Ithaca, NY). In the experiments, SLBs composed of a 1,2-dioleoylsn-glycero-3-phosphocholine (DOPC) lipid were initially formed using the solvent-assisted lipid bilayer (SALB) technique [40]. The SLBs were freshly prepared before each experiment. A baseline signal in the aqueous buffer solution (10 mM Tris, 150 mM NaCl, pH 7.5) was recorded, followed by an exchange relative to the isopropanol solution, an addition of 0.5 mg/mL of DOPC lipids in the isopropanol solution, and finally a solvent exchange with the PBS buffer solution to form the SLB. Subsequently, after the stabilization of a baseline signal in the aqueous Tris buffer solution, 0.1 mg/mL of bovine serum albumin (BSA) in an aqueous solution was added as a blocking agent to prevent nonspecific protein adsorptions onto the SLB-coated surface [57]. After completing the bilayer formation, the test compound in the PBS solution was added under a continuous flow rate. A washing step with the PBS solution completed the procedure. All liquid samples were added into the measurement chamber under continuous flow conditions using a peristaltic pump (Reglo Digital, Ismatec, Glattbrugg, Switzerland), and the flow rate was set at 50 μL/min. During the experiments, the temperature in the measurement cell was maintained at 25.0 ± 0.5 °C. Measurement data were collected at the third (*n* = 3), fifth (*n* = 5), and seventh (*n* = 7) overtones using the Q-Soft software program (Biolin Scientific). All presented data were collected at the fifth (*n* = 5) overtone. Data processing was performed in Q-Tools (Biolin Scientific) and OriginPro (OriginLab, Northampton, MA, USA) software programs.

### 3.5. Time-Lapse Fluorescence Microscopy

Epifluorescence microscopy experiments were conducted to directly observe real-time membrane morphological changes in the SLBs on glass surfaces upon treatment with LNA, LLA, and OA. The experiments were conducted using an Eclipse TI-E inverted microscope (Nikon, Tokyo, Japan) with a 60× magnification (NA = 1.49) oil-immersion objective lens (Nikon), and micrograph images were collected with an iXon 512 pixel × 512 pixel EMCCD camera (Andor Technology, Belfast, UK). The pixel size was 0.267 × 0.267 μm^2^. A fiber-coupled mercury lamp (Intensilight C-HGFIE, Nikon) was used to illuminate fluorescently labeled DOPC phospholipids (0.5 mol % Rh-PE) through a TRITC filter [58]. SLBs were fabricated using the solvent-assisted lipid bilayer (SALB) method [40,59] on a sticky slide VI 0.4 that was enclosed within a microfluidic flow-through chamber (Ibidi GmbH, Gräfelfing, Germany). After complete lipid bilayer formation, the measurement chamber was rinsed with a PBS buffer solution, and then the test compound was introduced to the chamber at a continuous flow rate of 50 μL/min. Time-lapse micrographs were recorded every 5 s for a total duration of 60 min at concentrations above CMC and below CMC at room temperature (21 °C). The initial time, t = 0 s, was defined by when the test compound was initially injected. Image analysis was conducted using the ImageJ software program (National Institutes of Health, Bethesda, MD, USA).

### 3.6. Statistical Analysis

All experiments were performed in triplicate unless otherwise stated. A two-tailed T-test was carried out to analyze the statistical significance of measurements by using Microsoft Excel (Microsoft, Redmond, WA, USA). Correlations with *p*-value < 0.05 were considered statistically significant (* *p* < 0.05, ** *p* < 0.01, and *** *p* < 0.001).

## 4. Conclusions

In this work, we investigated the real-time membrane remodeling effects of long-chain unsaturated fatty acids—linolenic acid (LNA, C18:3), linoleic (LLA, C18:2), and oleic acid (OA, C18:1)—by using model membranes and carrying out systematic biophysical measurements. LNA had a higher CMC value than other unsaturated fatty acids, indicating that the unsaturation degree affects the self-assembly property of fatty acids. Using the combination of QCM-D and fluorescence microscopy, we discovered that the fatty acids caused distinct changes in membrane morphology in a concentration-dependent manner, which was characterized by the development of tubules protruding from the lipid bilayer membrane. We also identified that the compounds induced membrane remodeling activity against SLBs at concentrations primarily above their respective CMC values. The results showed that LNA exhibited the highest membrane remodeling ability, suggesting that the fatty acids with a higher degree of unsaturation lead to significant changes in the membrane remodeling process but that they also required high concentrations to initiate remodeling due to increased CMC values. Overall, our findings provide insights into how the self-aggregation properties and degree of unsaturated bonds in long-chain unsaturated fatty acids can modulate membrane destabilization.

## Figures and Tables

**Figure 1 ijms-24-09639-f001:**
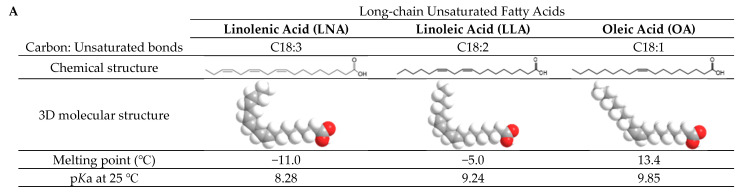
Overview of experimental strategy with materials used in this study. (**A**) Molecular structure and physicochemical properties of long-chain unsaturated fatty acids used in this study [26,27,28]. Melting points for LNA and LLA were taken from [27] and from [28] for OA. All p*K*a values were taken from [29]. (**B**) Schematic illustration of the experimental strategy to characterize how long-chain unsaturated fatty acids interact with lipid membranes via supported lipid bilayer (SLB) platforms.

**Figure 2 ijms-24-09639-f002:**
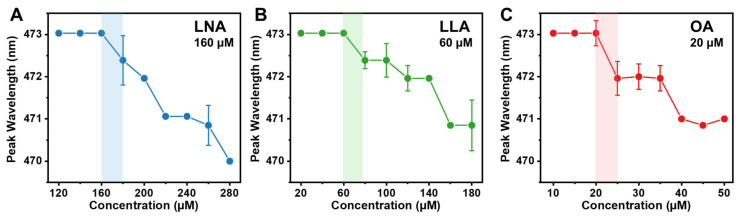
Determination of critical micelle concentration (CMC) values for LNA, LLA, and OA using fluorescence spectroscopy. Peak wavelength is shown as a function of compound concentration in the PBS buffer for (**A**) LNA, (**B**) LLA, and (**C**) OA. The CMC value is defined as the highest test concentration at which no peak shift occurs. Data are reported as mean ± standard deviation from six technical replicates (*n* = 6).

**Figure 3 ijms-24-09639-f003:**
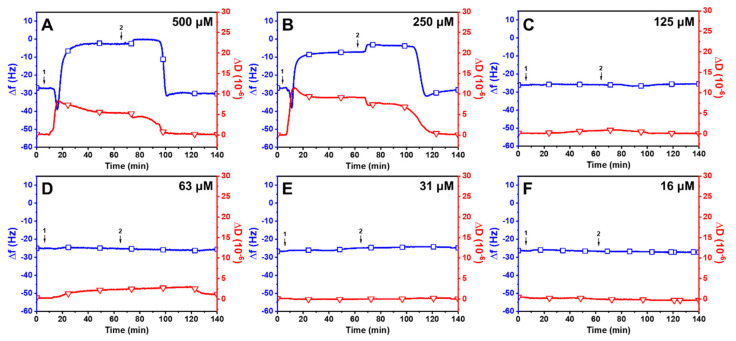
QCM-D investigation of concentration-dependent LNA treatments on SLBs at a pH of 7.5. Δ*f* (blue line with squares) and Δ*D* (red line with triangles) shifts are presented as a function of time for (**A**) 500 μM, (**B**) 250 μM, (**C**) 125 μM, (**D**) 63 μM, (**E**) 31 μM, and (**F**) 16 μM LNA. The initial baseline values recorded at t = 0 min indicate the formation of an SLB on the silica surface. LNA was introduced at t = 5 min (arrow 1), followed by a buffer washing step (arrow 2) after the measurement signals stabilized.

**Figure 4 ijms-24-09639-f004:**
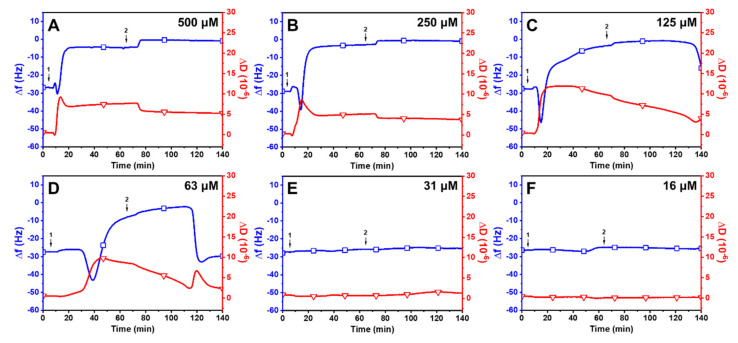
QCM-D investigation of concentration-dependent LLA treatment on SLBs at a pH of 7.5. Δ*f* (blue line with squares) and Δ*D* (red line with triangles) shifts are presented as a function of time for (**A**) 500 μM, (**B**) 250 μM, (**C**) 125 μM, (**D**) 63 μM, (**E**) 31 μM, and (**F**) 16 μM LLA. The initial baseline values recorded at t = 0 min indicate the formation of an SLB on the silica surface. LLA was introduced at t = 5 min (arrow 1), followed by a buffer washing step (arrow 2) after the measurement signals stabilized.

**Figure 5 ijms-24-09639-f005:**
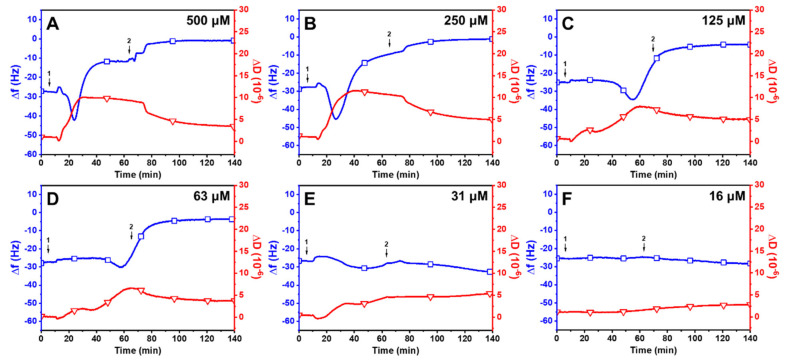
QCM-D investigation of the concentration-dependent OA treatment on SLBs at a pH of 7.5. Δ*f* (blue line with squares) and Δ*D* (red line with triangles) shifts are presented as a function of time for (**A**) 500 μM, (**B**) 250 μM, (**C**) 125 μM, (**D**) 63 μM, (**E**) 31 μM, and (**F**) 16 μM OA. The initial baseline values recorded at t = 0 min indicate the formation of an SLB on the silica surface. OA was introduced at t = 5 min (arrow 1), followed by a buffer washing step (arrow 2) after the measurement signals stabilized.

**Figure 6 ijms-24-09639-f006:**
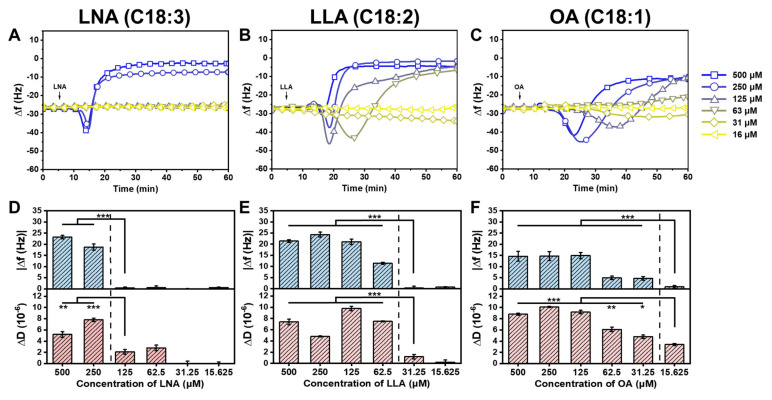
The trend in QCM-D measurement shifts for membrane remodeling behavior induced by different long-chain unsaturated fatty acids. Δ*f* shifts are presented as functions of time for (**A**) LNA, (**B**) LLA, and (**C**) OA at 16–500 uM concentrations until 60 min. Fatty acids were added at t = 5 min (arrow). Column graph of net |Δ*f*| shifts (upper panel) and Δ*D* shifts (lower panel) at 60 min for (**D**) LNA, (**E**) LLA, and (**F**) OA. Net |Δ*f*| and Δ*D* shifts are reported as | Δ*f*_measured_ − Δ*f*_bilayer_| and Δ*D*_measured_ − Δ*D*_bilayer_ shifts, respectively. The dotted line in the lower panel represents fatty acid concentrations above (to the left) and below (to the right) CMC ranges. Data are reported as the mean ± standard deviation from *n* = 3 measurements. The asterisk indicates a statistically significant difference between the initial concentration below CMC and the individual concentration above CMC values (* for *p*-value < 0.05, ** for *p*-value < 0.01, and *** for *p*-value < 0.001).

**Figure 7 ijms-24-09639-f007:**
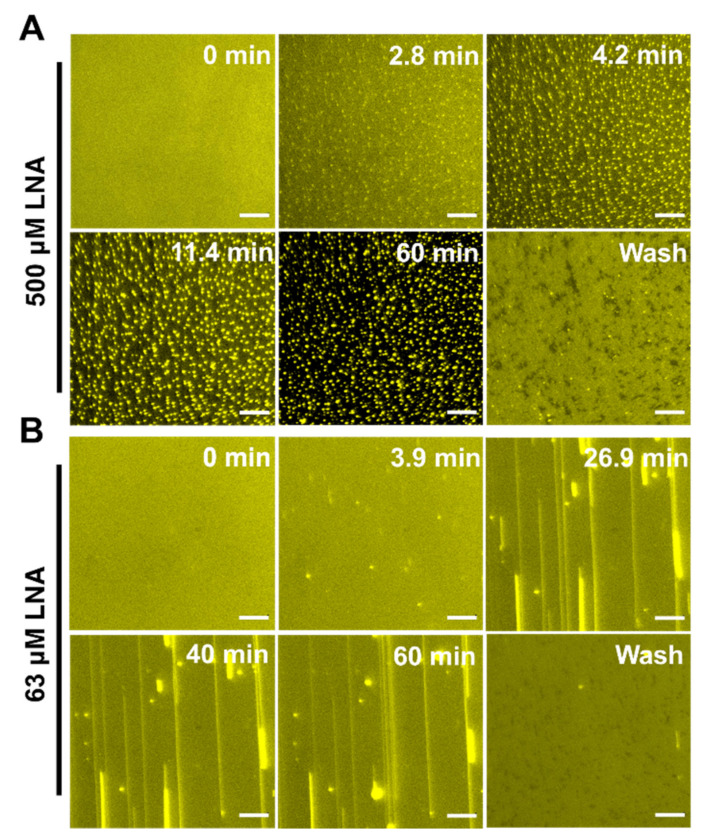
Time-lapse microscopic observation of LNA-induced membrane morphological responses on SLBs at concentrations above CMC and below CMC. (**A**) Image snapshots at various time points depict nucleation sites from which aggregates proliferate upon the treatment of SLB with 500 μM LNA at a pH of 7.5. (**B**) Image snapshots at various time points depict nucleation sites from which tubules grow upon the treatment of SLB with 250 μM LNA at a pH of 7.5. At t = 0 min, the LNA solution was added to the measurement chamber. The scale bar is 20 μm.

**Figure 8 ijms-24-09639-f008:**
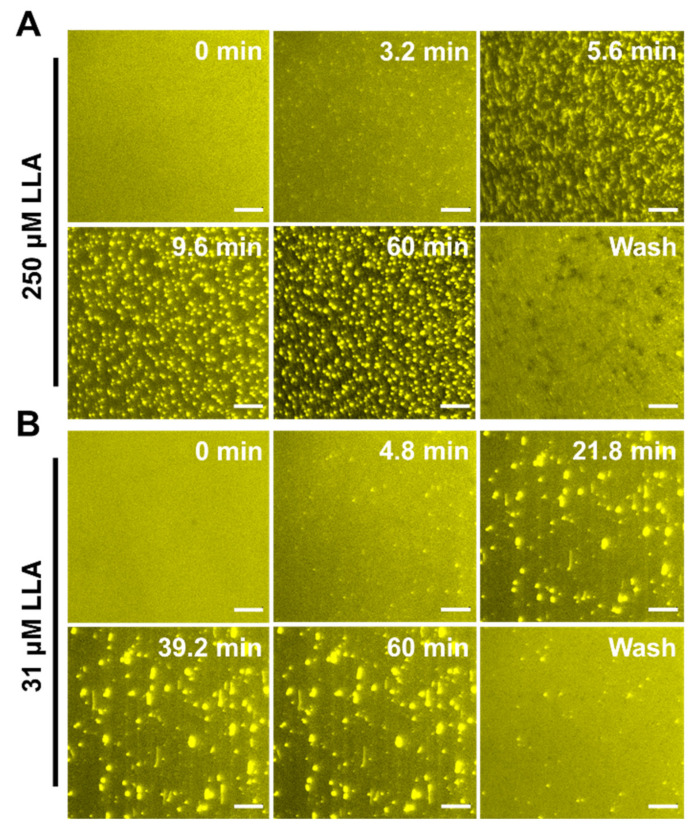
Time-lapse microscopic observation of LLA-induced membrane morphological responses on SLBs at concentrations above CMC and below CMC, respectively. (**A**) Image snapshots at various time points depict nucleation sites from which aggregates proliferate upon the treatment of SLB with 250 μM LLA at a pH of 7.5. (**B**) Image snapshots at various time points depict nucleation sites from which tubules grow upon the treatment of SLB with 31 μM LLA at a pH of 7.5. At t = 0 min, the LLA solution was added to the measurement chamber. The scale bar is 20 μm.

**Figure 9 ijms-24-09639-f009:**
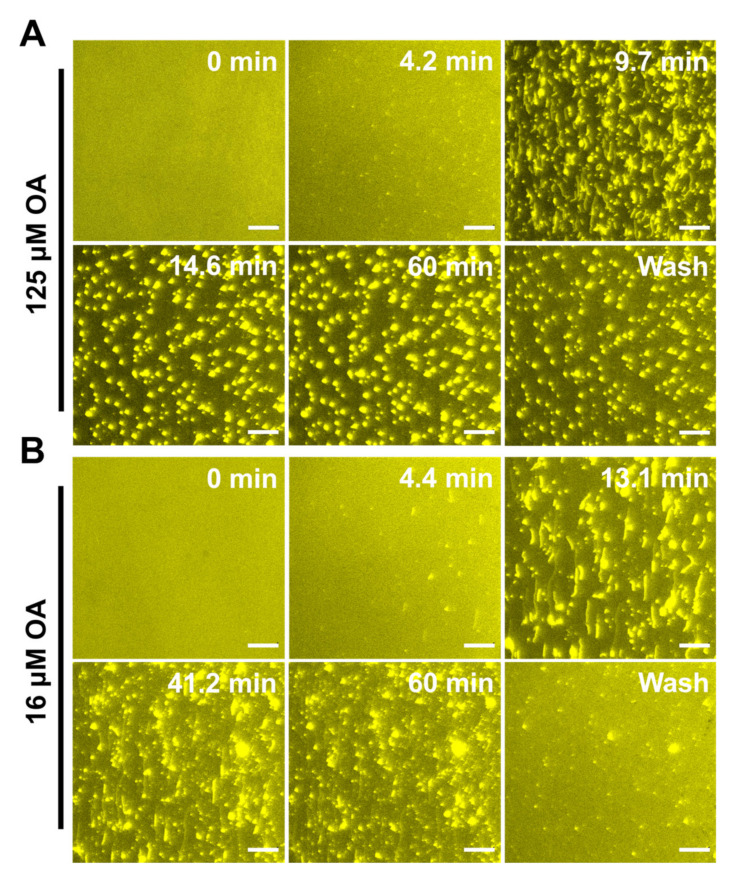
Time-lapse microscopic observation of OA-induced membrane morphological responses on SLBs at concentrations above CMC and below CMC. (**A**) Image snapshots at various time points depict nucleation sites from which aggregates proliferate upon the treatment of SLB with 125 μM OA at a pH of 7.5. (**B**) Image snapshots at various time points depict nucleation sites from which tubules grow upon the treatment of SLB with 16 μM OA at a pH of 7.5. At t = 0 min, the LLA solution was added to the measurement chamber. The scale bar is 20 μm.

## Data Availability

The data presented in this study are available upon request from the corresponding author.

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
