# Peer review of "Lipid Membrane Remodeling by the Micellar Aggregation of Long-Chain Unsaturated Fatty Acids for Sustainable Antimicrobial Strategies"

_ijms, 2023, doi:10.3390/ijms24119639_

Round 1

Reviewer 1 Report

In the current article (ijms-2378071), the authors describe the interaction of Natural Antimicrobial fatty acids with the supported lipid bilayers (SLBs). The current study claimed that self-aggregation and degree of unsaturation of fatty acids could destabilize bacterial membrane thus these properties could be helpful in developing antimicrobial strategies against bacteria. Though the article is of great interest for scientific community and health professionals but the following queries and comments may be addressed.

  • Comments:

    • In abstract section, the results should be presented in more quantitative manner.
    • Reference may be provided for the methods used in the study.
    • Positive and negatives controls of the experiments should be clearly defined in the methodology and result section.
    • Proper statistical analysis should be performed and reflected in the results to show the significance of the findings.
    •  Figure 9. B: the 16 Mm OA should be changed to 16 μM OA.

Well written but minor mistakes should be removed.

Reviewer 2 Report

Congratulations !

Reviewer 3 Report

The article by Shin et al. describes the interaction between supported lipid bilayers (SLBs) and between acid long-chain unsaturated fatty acids—linolenic acid, linoleic, and oleic acid- using quartz crystal microbalance-dissipation (QCM-D) and fluorescence microscopy .

The work is well written and result interesting to the scientific community.

The authors indicate that the fatty acids caused distinct changes in the morphology of BPS in a concentration-dependent manner, and that they also identified that the compounds induced membrane remodeling activity against SLB at concentrations higher than their respective CMC values.

These fatty acids have promising antimicrobial activity that could be related to the properties described in this publication. However, the authors do not indicate the MIC of these compounds in bacterial strains. Since the interaction of these compounds with the SLB occurs at concentrations higher than the CMC, it would be very interesting to compare these data with the MIC available in the literature or experimentally.

Round 2

Reviewer 3 Report

This referee appreciate the response of the authors.

This referee suggest the publication of this manuscript in IJMS in the present form.